# Matrix Metalloproteinases in Chronic Obstructive Pulmonary Disease

**DOI:** 10.3390/ijms24043786

**Published:** 2023-02-14

**Authors:** Maria-Elpida Christopoulou, Eleni Papakonstantinou, Daiana Stolz

**Affiliations:** 1Department of Pneumology, Medical Center, Faculty of Medicine, University of Freiburg, 79106 Freiburg, Germany; 2Clinic of Respiratory Medicine and Pulmonary Cell Research, University Hospital, 4031 Basel, Switzerland

**Keywords:** matrix metalloproteinases (MMPs), chronic obstructive pulmonary disease (COPD), tissue inhibitors of MMPs (TIMPs)

## Abstract

Matrix metalloproteinases (MMPs) are proteolytic enzymes that degrade proteins of the extracellular matrix and the basement membrane. Thus, these enzymes regulate airway remodeling, which is a major pathological feature of chronic obstructive pulmonary disease (COPD). Furthermore, proteolytic destruction in the lungs may lead to loss of elastin and the development of emphysema, which is associated with poor lung function in COPD patients. In this literature review, we describe and appraise evidence from the recent literature regarding the role of different MMPs in COPD, as well as how their activity is regulated by specific tissue inhibitors. Considering the importance of MMPs in COPD pathogenesis, we also discuss MMPs as potential targets for therapeutic intervention in COPD and present evidence from recent clinical trials in this regard.

## 1. Matrix Metalloproteinases (MMPs)

Extracellular matrix (ECM) metalloproteinases (MMPs), or matrixins, are the major proteases in mammals. MMPs, together with astacins, ADAMs/adamalysins and seralysins, belong to the family of zinc endopeptidases collectively referred to as metzicins [1]. Their common feature is a highly conserved HEXXHXXGXXH(H/D) domain containing three histidines that bind to Zn^2+^ at the catalytic site and a conserved methionine sequence (XBMX) on the carboxyl side of the active site, forming a Met-turn, which is responsible for their final conformation [2,3,4,5].

MMPs are proteolytic enzymes characterized by their ability to degrade ECM proteins and the basement membrane. For each of the above components, there is at least one enzyme of the MMP family capable of degrading it. MMPs play an important role in regulating remodeling of the ECM to facilitate immune cell activity and regulate cell behavior by influencing biochemical and physical cues [6,7,8,9]. Thus, MMPs are involved in physiological processes such as morphogenesis, tissue regeneration, wound healing and angiogenesis [10,11], as well as in pathological processes such as osteolysis, arthritis/osteoarthritis, invasive cancer and fibrotic diseases [12,13,14].

In vertebrates, there are 28 different MMPs; however, only 23 MMPs are expressed in human tissues that are synthesized as pre-proenzymes [15,16]. Their architecture consists of an N-terminal signal peptide with variable length, a latency-maintaining pro domain, a catalytic domain with a Zn^2+^, a linker-“hinge” region and a C-terminal domain [16,17] (Figure 1). In all MMPs, apart from MMP-7 and MMP-26, there is a hemopexin-like domain at the C-terminus, which is essential for some of their actions (e.g., collagen cleavage). Additionally, in transmembrane MMPs, there is a transmembrane domain (TM) and a short cytoplasmic domain or a glycosylphosphatidylinositol (GPI) anchor that binds them to the cell surface. The structure of MMP-2 and MMP-9 differs from that of other MMPs, as in their catalytic domain, they contain cysteine repeats that resemble collagen binding sites of type II fibronectin [18,19] (Figure 1).

There are several ways to categorize MMPs. According to bioinformatic analysis, MMPs can be subdivided in five categories [16]: (1) MMPs anchored to the cellular membrane by a C-terminal GPI moiety (MMP-11, MMP-17 and MMP-25), (2) MMPs with a transmembrane domain (MMP-14, MMP-15, MMP-16 and MMP-24), (3) MMPs with three fibronectin-like inserts in the catalytic domain (MMP-2 and MMP-9), (4) non-furin-regulated MMPs (MMP-1, MMP-3, MMP-7, MMP-8, MMP-10, MMP-12, MMP-13, MMP-20 and MMP-27) and (5) all other MMPs (MMP-19, MMP-21, MMP-23, MMP-26 and MMP-28). Based on their sequential similarity and domain organization shown in Figure 1, as well as by their substrate specificity, MMPs are classified into six groups: collagenases, stromalysins, matrilysins, gelatinases, membrane-type MMPs and other MMPs (Table 1). 

Traditionally, substrate sequencing is viewed as the guiding principle for protease specificity. However, it has been demonstrated that the substrate specificity of MMPs is not guided by sequencing alone, but there are other substrate features, such as local triple-helix instability, possibly in combination with subtle variations in sequence, that may influence MMP–substrate specificity that needs to be further investigated [20]. For example, it has been shown that MMP-1 hydrolyzes type III collagen more rapidly than type I. MMP-2, which is primarily a gelatinase, can act like collagenase, albeit in a weaker manner [21]. MMP-8 and MT1-MMP (MMP-14) show a slight preference for type I collagen compared with type III. Type V collagen is hydrolyzed by MMP-2 and MMP-9 but not MMP-1, MMP-8 or MMP-13, whereas type XI collagen is cleaved by MMP-1, MMP-2 and MMP-9 but not by MMP-13 [20].

## 2. Regulation of MMP Activity

MMPs are expressed by a variety of cell types, such as macrophages, fibroblasts, endothelial cells, keratinocytes, etc., in response to hormones, cytokines and developmental factors. However, MMP-2 and MT-1 MMP are ubiquitous in a variety of tissues, and unlike other MMPs, they are constitutively expressed [22]. Because MMPs are critical enzymes involved in a variety of cellular processes, their activity is tightly controlled at the level of transcription and pro-peptide activation and by tissue inhibitors of MMPs (TIMPs). 

The expression of MMPs, apart from polymorphisms, in the promoter of MMP gene [23], is also regulated by epigenetic modifications at the level of transcription [24,25]. The promoters of at least 14 MMPs and those of all TIMPs contain CpG islands prone to methylation. The methylation level of certain CpG sites has been associated with the level of transcription of MMPs and correlated with the incidence and severity of inflammatory diseases such as COPD and asthma [26,27]. Additionally, transcriptional control has been observed via histone modifications and chromatin remodeling, while at the post-transcriptional level, several microRNAs (miRNAs) have been implicated [28,29]. For example, in COPD, significantly lower expression of miR-452 has been linked to increased expression of MMP-12, which is known to be an important effector of smoking-related diseases [30,31]. Experimental antagonism of miR-452 in differentiated monocytic cells resulted in increased expression of MMP-12 [31].

The next level of regulation of MMP activity is post-translational transformation from pre-proMMP to their active form [32]. It is noteworthy that most MMPs are not stored in the producing cells but are secreted immediately after their synthesis as latent zymogens, which are activated in the extracellular environment by proteolytic removal of the pro peptide [16]. After secretion, they can also accumulate in the cells through re-entry, where they interact with intracellular molecules to regulate signaling processes [33]. Intracellular MMPs are activated through oxidative stress, which induces conformational changes that activate MMPs even in the presence of the pro peptide, post-translational modifications by phosphorylation and proteolytic cleavage by furins [21]. There are also membrane-type MMPs (MT-MMPs) that are anchored to the cell surface via a transmembrane domain, a GPI or an N-terminal signal anchor [21]. MT-MMPs undergo intracellular activation by furin-like convertases, then proceed to the cell surface, where they can cleave and activate other pro-MMPs. MT-1 MMP dimer interacts with TIMP-2 to activate pro-MMP-2 on the cell surface [34].

MMP activity is further controlled by mechanisms such as compartmentalization (Figure 2). MMPs are classically viewed as extracellularly localized, but they have been found in every cellular compartment interacting with other proteins, proteoglycan core proteins and/or their glycosaminoglycan chains, as well as with other molecules [33,35,36]. They may also be subjected to endocytosis following secretion by LDL-related protein 1 receptor binding for MMP-2, MMP-9 and MMP-13 and by caveolae for MT-1 MMP [37]. In plasma, MMP activity is inhibited by α2-macroglobulin (A2M) [38]. Finally, glycosylation and other post-translational modifications may also affect the activity of MMPs, as well as their localization and their interaction with substrates and other proteins [32].

The enzymatic action of activated MMPs in tissues is regulated by endogenous TIMPs [39]. TIMPs are produced by the same cells that produce MMPs; in addition to inhibiting active MMPs, they can also behave as growth factors in specific cell types to mediate cell signaling [40]. To date, four members of the family have been confirmed: TIMP-1, TIMP-2, TIMP-3 and TIMP-4 [14,41]. All mammalian TIMPs consist of two distinct domains: an N-terminal domain of about 135 amino acids and a C-terminal domain of about 65 residues; they show great similarities in the amino acid sequence [41,42]. 

The presence of Cys at specific positions in the molecule results in the development of disulfide bonds, which are responsible for both their tertiary structure and molecular stability [4,43]. TIMPs bind non-covalently to MMPs in a 1:1 ratio and regulate their activity. Through their N-terminal tail, they bind to the active site of MMPs, rendering them inactive [39]. Of most importance is the integrity of the TIMP molecules, as even partial proteolysis renders them ineffective. However, in the process of complexation with MMPs, TIMPs may interact with several binding sites on MMPs, as well as the catalytic domain (CAT), forming an inhibitory complex. Although in the process of complexation with MMPs, the N-terminal region binds the CAT of the enzymes, it is believed that an interaction between C-terminal end regions of TIMPs and MMPs occurs first [44]. Moreover, the C terminus of TIMPs may interact with multiple binding sites of MMPs, such as the hemopexin domain (HPX), to regulate the functional properties of MMPs [45].

TIMPs play a significant role in inflammatory and cardiovascular diseases, as well as in cancer, as they can directly modulate ECM turnover and cell behavior [15]. The most well-studied inhibitors are TIMP-1 and TIMP-2. TIMP-1 is a mannose-rich glycoprotein with a molecular mass of 23 kDa, while TIMP-2, with a molecular mass of 22 kDa, is not glycosylated. TIMP-4 is a 22 kDa amino acid polypeptide with several N-glycosylated sites [46]. The action of TIMP-3, which has a molecular mass of 21 kDa, is not limited to the dissection of MMPs. TIMP-3 exhibits inhibitory action on a wide spectrum of substrates (e.g., proteins in the extracellular space), and high levels of TIMP-3 may affect the surfaceome, identifying deregulated molecular pathways [47,48]. TIMP-1, TIMP-2 and TIMP-3 interact with low-density lipoprotein receptor-related protein 1 (LRP-1), leading to TIMP endocytosis. Sulfated glycosaminoglycans antagonize the binding of LRP-1 to TIMP-3, thus blocking the endocytosis of the enzyme and affecting its activity at high GAG concentrations [47,49,50]. TIMP-1, TIMP-2 and TIMP-4 are secreted, whereas TIMP-3 is associated with the ECM [41,51]. Therefore, the four TIMPs vary in their regulation, affinity and mechanism of action. 

## 3. MMPs in COPD

Chronic obstructive pulmonary disease (COPD) is one of the main causes of human mortality globally. It is a chronic inflammatory disease characterized by structural remodeling of the airways and alveolar destruction. Alveolar damage is the result of excessive, uncontrolled and persistent proteolysis, leading to the degradation of selective components of the ECM. Several studies have shown that loss of elastin rather than loss of fibrillar collagen due to proteolytic destruction is the cause of the disease and the development of emphysema [52,53,54]. Elastolysis caused by specific MMPs produced by macrophages leads to poor lung function in COPD patients [55,56].

Cigarette smoke (CS) exposure is the most commonly encountered risk factor for COPD. Chronic smoking is associated with continuous recruitment of inflammatory cells and release of inflammatory mediators, such as MMPs, neutrophil elastase, chemokines, cytokines and reactive oxygen species. Epithelial cells and macrophages activate fibroblasts by releasing mediators, such as TGFβ, leading to airway remodeling. In addition, CS impairs structural cell function and initiates the EMT, a process that leads to endothelial cell dysfunction, which hampers tissue repair and eventually leads to fibrosis [57].

The number of macrophages appears to be upregulated in lungs of smokers compared to non-smokers [58], while increased macrophage influx has been associated with COPD severity [59]. However, the exact role of macrophages in COPD remains elusive due to their functional heterogeneity [60]. M1 macrophage phenotype seems rather harmful, contributing to ECM deposition by producing profibrotic cytokines that promote myofibroblast formation [61,62], while M2 macrophages provide benefits by clearing excess ECM deposition [63]. Nonetheless, in long-term inflammatory conditions, such as the causatives of COPD, macrophages promote persistent ECM degradation. For example, macrophages in bronchoalveolar lavage (BAL) of COPD smokers are more prone to degrade elastin than macrophages from healthy individuals due to increased MMP activity [56,64]. Accordingly, in several studies in humans and mice, MMPs, particularly macrophage-derived MMPs, have been associated with the pathogenesis of COPD. MMPs (Table 1) were shown to promote inflammation and disease progression by influencing macrophage activation rather than acting directly to degrade elastin [56,65]. The primary role of MMPs is to degrade the ECM; however, they can act on many more substrates than the ECM and have multiple modes of action. Therefore, MMPs can lead to alveolar destruction either by directly degrading the ECM or indirectly by tuning the proteolytic phenotype of macrophages, as may be the role of MMP-10 and MMP-28 [33]. Furthermore, MMPs, through their exosites, bind to different macromolecules and are able to control cellular activities without functioning as proteinases, as demonstrated for MMP-12 [33]. 

Regulation of MMP production in COPD may occur at a transcriptional level; it has been shown that several proteins, including early-growth response gene product 1 (EGR1), nuclear factor kappa B (NF*κ*B), globin transcription factor 1 (GATA1), activator protein 1 family members (AP-1) and signal transducer and activator of transcription 3 (STAT3C), affect the MMP gene family. CS extract induced Egr-1 protein expression and increased Egr-1 DNA-binding activity in human lung fibroblasts [66]. Treatment with a mixture of tumor necrosis factor (TNF)-*α*, interleukin (IL)-1*β* and interferon (IFN)-*γ* resulted in an increase in the activity of MMP-2 in lung fibroblasts from EGR1 control (+/+) mice but was not detected in that of EGR1 null (−/−) mice, whereas MMP-9 was regulated by EGR1 in a reverse manner [66]. 

A summary of the role of MMPs in COPD is shown in Table 2.

### 3.1. Collagenases in COPD

#### 3.1.1. MMP-1

Matrix metalloproteinase-1 (MMP-1) is a collagenase that degrades collagen, which is significantly associated with COPD [67] and is a potential biomarker to better understand the course of COPD in patients [68]. Serum and plasma levels of MMP-1 were found to be elevated in COPD patients and correlated with COPD severity, whereas serum MMP-1 levels were found to be significantly elevated in smokers [69]. Sputum analysis showed increased MMP-1 levels in both smokers and patients with more advanced COPD [70]. In addition, a functional variant of MMP-1, rs1799750 G/GG, was associated with a high risk of COPD [71].

#### 3.1.2. MMP-8

MMP-8 is a metalloendopeptidase localized in neutrophils and macrophages known to be involved in COPD [72]. Enhanced levels of MMP-8 were identified in induced sputum of COPD patients [73]. In addition, PBMCs and plasma from COPD patients showed high levels of mRNA and protein expression for MMP-8, respectively, and expression was higher during exacerbations compared to steady state, suggesting a role of MMP-8 in COPD [74,75]. MMP-8 protein, although suppressed, was also detected in exhaled breath condensate of COPD patients, suggesting its association with excessive inflammation [74]. In mice deficient in MMP-8, there is increased neutrophilic inflammation in the BAL and peri bronchial region [76], whereas serum levels of MMP-8 in patients with atopic COPD were significantly higher than those determined in patients with non-atopic COPD [77]. Thus, MMP-8 mediates the inflammatory response by potentially inducing neutrophil apoptosis, making COPD patients more susceptible to acute exacerbation due to the effect of allergens [78]. Furthermore, MMP-8 levels were associated with acute dyspnea attacks in patients with atopic COPD, suggesting that MMP-8 may be a potential advisory tool for clinical practice [77].

#### 3.1.3. MMP-13

MMP-13, which is both a collagenase and an elastase, is another critical metalloproteinase in lung destruction during the development of COPD [79]. Collagenase-3 levels were found to be upregulated in mice after long-term exposure to cigarette smoke and in patients with COPD [72,80,81]. MMP-13 expression in α1,6-fucosyltransferase-deficient mice and Zntb7 knockout mice was associated with the development of airspace enlargement and therefore with an emphysema-like phenotype [82,83]. In humans, MMP-13 has been shown to be involved in COPD exacerbations, as its levels remain elevated in smokers after viral infections [84]. Furthermore, MMP-13-mediated cleavage of α-1 antitrypsin has been reported to reduce MMP-13 activity and protect against lung damage [85]. Therefore, targeting MMP-13 through specific inhibitors or AAT therapies may be beneficial for the treatment of COPD [86]. 

### 3.2. Stromalysins in COPD

#### 3.2.1. MMP-3

MMP-3 levels have been found to be elevated in patients with COPD, particularly among patients carrying the 6A6A genotype. High levels of MMP-3 are thought to potentially contribute to excessive ECM degradation and worse lung function [87]. Interestingly, although smoking has been widely associated with upregulation of MMPs in serum, MMP-3 levels were not found to be elevated in smokers [69]. On the other hand, in another study, MMP-3 concentration in BAL was associated with CT markers of small airway disease and appeared to be related to the severity of emphysema [88]. Further studies need to be conducted to determine the role of MMP-3 in the pathogenesis of COPD in terms of risk and severity. 

#### 3.2.2. MMP-10

MMP-10 is expressed by macrophages and CD68-positive cells in the lungs and, to a lesser extent, by epithelial cells in response to acute inflammatory conditions [56,89]. Macrophage-derived MMP-10 mitigates the proinflammatory response by controlling macrophage activation through restraint of M1 polarization and promoting the ability of M2 macrophages to control the expression of gelatinolytic MMPs, particularly MMP-13 [90,91]. 

Smoking and tobacco consumption have been shown to affect MMP-10 levels, and MMP-10 has been found to be elevated in the lungs of COPD patients [69]. More importantly, its expression is associated with small airway disease and the severity of emphysema [88]. Furthermore, MMP-10 in mice appeared to contribute to the development of cigarette-smoke-induced disease by directing macrophage–ECM macrophage remodeling [83]. Thus, MMP-10 appears to play an important role within the ECM assembly, suggesting that this proteinase is a relevant target for disease control. However, the lack of a selective inhibitor has prevented researchers from understanding whether targeting MMP-10 is favorable for the treatment of COPD. A potent inhibitor with no obvious zinc-binding moiety was developed that can simultaneously inhibit MMP-10 and MMP-13 [4,92]. Still, main goal of researchers remains to obtain an inhibitor that selectively inhibits MMP-10 against its close counterparts.

### 3.3. Matrilysins in COPD

#### MMP-7

MMP-7, or matrilysin, is a potent elastase in humans but not in mice that is expressed by the mucosal epithelium and macrophages in humans but only by epithelial cells in mice [56,93]. MMP-7 expression has been found to be increased in the blood of COPD patients compared to healthy subjects and is associated with deterioration of lung function [94]. In addition, an SNP in the promoter of the MMP-7 gene (rs1156818) is associated with a high risk of COPD [95]. In another study, MMP-7 serum levels were reported to be elevated in patients with emphysema and correlated with GOLD stages [67,69].

### 3.4. Gelatinases in COPD

#### 3.4.1. MMP-2

Serum levels of MMP-2 (gelatinase A) are significantly elevated in stable COPD patients compared with asthmatic patients and controls, suggesting MMP-2 as a potential biomarker for COPD [96]. These data were also confirmed by mass spectrometric analysis of the COPD proteome, where a positive fold change in MMP-2 expression was detected [96]. In another study, local expression of MMP-2, as well as MMP-9 and TIMP-1, was associated with pathological changes in the pulmonary interstitium and lung function of COPD patients, suggesting their involvement in COPD progression [97]. Although MMP-2 levels are elevated in patients with COPD, the functional role of MMP-2 in COPD has not yet been established. However, it was discovered that MMP-2 expression and activity were significantly increased in lung tissues in humans after injury and in rats with pulmonary fibrosis [98,99]. Therefore, considering the increased expression of MMP-2 and TIMP-1 in COPD patients and their association with collagen and elastic fiber formation, MMP-2 appears to be involved in ECM remodeling and interstitial fibrosis and to be a suitable therapeutic target for COPD pathogenesis [100].

#### 3.4.2. MMP-9

MMP-9 (gelatinase B) is an elastase expressed in both mice and humans. Although its expression in COPD is increased [101], its role remains questionable, mainly due to controversial data on its involvement in COPD. In mice, the absence of MMP-9 did not protect them from developing emphysema in response to LPS-induced inflammation [102]. On the other hand, transgenic overexpression of MMP-9 in macrophages led to impulsive emphysema in mice [101]. In another transgenic model, MMP-9 deficiency protected against alveolar expansion in response to IL-13-induced inflammation [103]. However, MMP-9 may have triggered IL-13-mediated alveolar remodeling. In COPD patients, MMP-9 expression did not vary in different lung compartments, and there was no association either between increased blood MMP-9 expression and progression of emphysema or between MMP-9 mRNA levels in macrophages and markers of ongoing lung injury [104]. However, a polymorphism of MMP-9, C1562T, was associated with disease susceptibility in middle-aged and elderly people [105].

It was shown that COPD patients are prone to produce higher levels of MMP-9 and MMP-9/TIMP-1 complex than healthy individuals, which is correlated with decreased FEV1% levels in patients with COPD compared to controls [106]. Moreover, the levels of MMP-9 and the ratio with TIMP1 have been associated with increased risk of death [107]. Increased levels of MMP-9, TIMP-1 and TIMP-2 were also observed in BAL during acute exacerbations of COPD and were negatively correlated with predicted FEV1%, indicating that MMP-9 and TIMPs may be persistent aggravating factors associated with airway remodeling and obstruction, suggesting a pathway connecting frequent exacerbations to lung function decline [108]. MMP-9 in COPD is linked to inflammation and lung remodeling, as it uniquely mediates pulmonary inflammation through ECM degradation, neutrophil chemotaxis and augmentation of inflammation. Elevated MMP-9 was independently associated with the risk of acute exacerbations in COPD in two well-characterized COPD cohorts of the SPIROMICS and COPDGene studies, indicating that MMP-9 may serve as a prognostic biomarker and potential therapeutic target in COPD [109]. In a recent study, MMP-9 serum levels, along with PGE2 and COX-2 levels, were found to be enhanced in COPD patients relative to healthy subjects and correlated with GOLD grade, CAT score and clinical history [110]. Furthermore, MMP-9 could also serve as a therapeutic target in COPD, as novel interventions targeting MMP-9 modulation are being investigated. In human and mouse models, cigarette smoke has been shown to enhance MMP-9 production [111] via p38 MAPK/ERK [102] and RANKL [112], respectively. These findings elucidate the molecular mechanisms involved in MMP-9 induction in COPD and suggest potential new targets for intervention.

### 3.5. Membrane-Type MMPs in COPD

#### MMP-14

It has been shown that cigarette smoke and tobacco smoke extract (TSE) can promote the secretion of extracellular vehicles (EVs) by both macrophages and bronchial epithelial cells, which contribute to the release of MMP-14 [113]. Thus, MMP-14 may be involved in emphysema [114]. This view is supported by the increased MMP-14 activity and protein found in the airway epithelium in a mouse model exposed to cigarette smoke [115]. However, decreased MMP-14 activity and protein reduce transcripts of mucin 5AC that play an important role in the development of COPD [116] Although Mmp14 −/− mice have an emphysema-like phenotype, this was not associated with abnormalities in collagen and elastin deposition or increased inflammation [117]. Therefore, the role of MMP-14 in COPD remains unclear.

### 3.6. Other MMPs in COPD

#### 3.6.1. MMP-12

MMP-12 has been shown to have a proinflammatory function since after its secretion, it is internalized and translocated to the nucleus, leading to enhanced NF-kB signaling [118,119]. In addition, its non-catalytic C-terminal end has potent antibacterial activity [120]. However, the role of these non-proteolytic actions of MMP-12 has not been correlated with COPD. Nevertheless, a number of studies support the involvement of MMP-12 in alveolar destruction and the pathogenesis of COPD, acting either as an elastase or affecting macrophage activation [121]. The expression levels of MMP-12 were found to be enhanced in BAL, in peripheral blood mononuclear cells (PBMCs) and in serum from patients with COPD compared to healthy subjects [46,88], while there is plenty of data supporting that MMP-12 is required for the development of emphysema in mice [122]. On the other hand, the presence of single-nucleotide polymorphisms in the promoter of MMP-12, such as the (-82) A/G allele of SNP rs2276109 or Asn357Ser (A/G) of rs652438, has been associated with reduced risk of developing COPD and better prognosis [46]. However, other studies have associated these SNPs with severe and very severe COPD (GOLD stages III and IV) [123]. In mouse models exposed to cigarette smoke, it has been proposed that macrophage influx is dependent on MMP-12, indicating that MMP-12 acts on elastin degradation [33]. This notion is supported by the fact that MMP-12 is required for the generation of a potent macrophage chemoattractant, which is a six-amino-acid fragment of elastin-VGVAPG [124,125]. On the other hand, it is tempting to hypothesize that this proteinase affects macrophage activation by regulating the proinflammatory activity of macrophages, as well as their ability to express other MMPs that may be implicated in ECM degradation.

#### 3.6.2. MMP-28

MMP-28, like MMP-10, plays a causative role in cigarette-smoke-induced emphysema [56]. The most recently discovered MMP is constitutively expressed in many human and animal tissues in the epithelium, as well as in monocytes and macrophages. In addition, it is expressed in human COPD lung tissue, while cigarette-smoke-exposed mice have been shown to present increased MMP-28 mRNA levels in both alveolar lung tissue and macrophages [126]. This suggests a role and contribution of MMP-18 in emphysema. MMP-28 is neither an elastase nor a matrix-degrading proteinase. Nevertheless, MMP-28 influences the inflammatory response by affecting the inflammatory activity of macrophages. In contrast to MMP-10, which promotes a shift in the M1 phenotype to the M2 phenotype of macrophages with subsequent functional changes, MMP-28 is associated with stimulation of chemokine expression [56]. Thus, MMP-28 appears to promote the inflammatory response in the pathogenesis of emphysema, although its specific contribution and mode of action remain to be elucidated.

## 4. TIMPs in COPD

Chronic inflammation is one of the main aspects of COPD pathogenesis leading to excessive remodeling of the extracellular space and inevitable deterioration of airflow function [127]. In addition to genetic factors, which account for only 1–5% of COPD patients, proteases and their specific inhibitors play a crucial role in pathogenesis [79,128], while the balance between MMPs and TIMPs is essential for ECM homeostasis [15]. AS mentioned above TIMPs are a group of low-weight glycoproteins that regulate the activity of MMPs by binding the zinc ion to the catalytic center of MMPs to form stable complexes. However, increased inflammation or demand in remodeling activities may cause an imbalance of MMP and TIMP levels [39]. TIMPs are expressed by a variety of cells in the lungs, including macrophages, activated fibroblasts, and airway smooth muscle and epithelial cells [129]. Whether they are secreted into the ECM in soluble form or localized to the cell membrane through interaction with other proteins, they exert a direct or indirect effect on the regulation of ECM turnover [130].

### 4.1. TIMP-1

TIMP-1 is a critical regulator of extracellular matrix degradation [131] and plays a key role in limiting inflammation after injury [132,133]. Impaired protease–anti-protease imbalance in COPD patients is associated with the presence of airway injury [79,134,135,136,137]. In particular, TIMP-1 has the ability to moderate ECM degradation both in healthy tissues and under pathological conditions [40,42,138,139]. For example, the presence of circulating TIMP-1 has been shown to control proteolysis in the cardiovascular system [140,141]. Furthermore, TIMP-1 preferentially inhibits MMP-9, a key metalloproteinase in the pathogenesis of emphysema; therefore, TIMP-1 may exert protection, while the levels of MMP-9 and the MMP-9/TIMP-1 ratio are reliable predictors of emphysema [142,143]. On the other hand, TIMP-1 levels were found to be elevated in COPD patients, and plasma TIMP-1 levels were associated with disease severity [106,135,138]. Furthermore, TIMP-1 was found to be enhanced in PH, a very common and lethal comorbidity of COPD, compared to COPD patients, suggesting that TIMP-1 levels could be used as a biomarker to identify high-risk patients [138].

### 4.2. TIMP-2

TIMP-2 is known to be involved in abnormal ECM accumulation either directly or indirectly through inhibition of MMP-14 or activation of MMP-2 [144,145,146]. TIMP-2 in COPD subjects was found in the bronchial epithelium and connective tissue, along with apparently large numbers of immunoreactive TIMP-2 endothelial cells [146]. In another study, Tacheva T. et al. showed that plasma levels of TIMP-2 in COPD patients were higher compared to controls with levels correlated with plasma MMP-2 levels [147]. In addition, the MMP-2/TIMP-2 ratio was elevated in COPD patients and representative of tobacco use [147]. Although TIMP-2 may be a potential biomarker for COPD, it cannot be correlated with disease severity.

### 4.3. TIMP-3 

TIMP-3 has many different functions in terms of regulating inflammation. TIMP-3 is a major inhibitor of ECM remodeling; however, it has been found to mediate fibrosis [47]. In a study comparing tissues from healthy control subjects with severe cases of tissue remodeling caused by diseases such as COPD GOLD stage IV and end-stage IPF, TIMP-3 levels were found to be significantly elevated [148]. Hence, TIMP-3 was identified as a disease regulator for COPD, strongly influencing tissue homeostasis.

### 4.4. TIMP-4

There is a lack of existing literature on TIMP-4; nevertheless, some data support its dual role in both protecting the ECM from proteolysis and limiting fibrosis [149,150]. In COPD, TIMP-4 protein levels have been found to be significantly upregulated in the serum of patients, along with increased expression of TIMP-4 mRNA in PBMCs [46]. Although these data demonstrate the involvement of TIMP-4 in COPD, they fail to correlate its expression with pulmonary lung function and to unveil its clinical significance. Therefore, further studies should be conducted to clarify the role of TIMP-4 in the pathogenesis of COPD.

## 5. MMPs as Targets for Therapeutic Intervention in COPD

Currently, there are only a few disease-modifying therapies for COPD treatment. Considering the importance of MMPs in COPD pathogenesis, the development of agents that could target the action of MMPs may have beneficial effects on disease progression [151,152]. In recent years, many studies have been conducted focusing on the discovery of small molecules as potent and selective inhibitors of MMPs [153,154]. Previous studies have shown that simvastatin modulates the expression of MMP-2 and MMP-9 in lung cancer tissue [155], as well as in a human lung adenocarcinoma cell line [156], indicating that simvastatin may play a role in in the prevention and treatment of lung cancer. As mentioned above, MMP-12 regulates the inflammatory response in mice, and its expression has been associated with disease severity in humans. Preclinical studies in COPD and emphysematous lungs support the notion that targeting MMP-12 could be a very promising therapeutic approach [122,157]. A specific inhibitor of MMP-9/MMP-12, AZ11557272, has been shown to protect mice against an increase in small airway thickness. When tested in smoke-exposed guinea pigs, the same inhibitor increased total lung capacity, residual volume and vital capacity [81,158]. Furthermore, AS111793, a specific inhibitor of MMP-12 has been shown to reduce the inflammatory processes associated with cigarette smoke exposure in mice [159]. Another selective inhibitor of MMP-12, MMP-408, has been shown to block rhMMP-12-induced lung inflammation in a mouse model [160]. Furthermore, mice exposed to porcine pancreatic elastase but treated with MMP-408 exhibited a significant decrease in emphysema-like pathology compared to vehicle-treated mice [161]. In addition, the dual MMP-9/MMP-12 inhibitor AZD1236 has been tested in phase II clinical trials, reaching permissible safety levels, although its therapeutic potential has not been established [162]. The endolysosomal mucolipid cation channel 3 (TRPML3) has been proposed as a potential drug for the treatment of COPD and emphysema. TRPML3 is uniquely expressed in alveolar macrophages and thus regulates the clearance of MMP-12 [163]. In another study, two agents with single-digit nanomolar affinity (compounds 25 and 26) resembling the structure of the proteasome inhibitor carfilzomib exhibited selectivity for MMP-12 [161]. Compounds 25 and 26 led to an improved emphysema phenotype in mice compared to those treated with vehicle; therefore they can be considered as potential novel therapeutic agents for the treatment of COPD (Figure 3).

## 6. Future Perspectives

Given the proven role of MMPs in the pathophysiology of COPD it is of utmost importance to identify prognostic or diagnostic indicators for the risk, prevalence and stage of MMP-induced COPD. Such an approach would require large clinical trials including cohorts of well-characterized COPD patients and assessments of MMPs in sputum, BAL, blood and in tissues in association with demographic characteristics, as well as results from questionnaires, clinical assessments and quantitative CT (QCT). The results from such studies may reveal novel targets for interventions targeting MMP-associated pathways.

As COPD is a disease triggered by various risk factors, the Lancet Commission towards the elimination of COPD has recently suggested that the disease should be classified into five types on the basis of the predominant risk factor driving the disease: (1) genetics, (2) early-life events, (3) respiratory infections, (4) tobacco exposure or (5) other environmental exposures [164]. Investigation of MMPs in large cohorts including patients from all the above types would highlight the impact of MMPs/TIMPs in pathophysiological mechanisms related to each of these risk factors, which could be translated into distinct diagnostic, prognostic and therapeutic considerations (Figure 3).

## Figures and Tables

**Figure 1 ijms-24-03786-f001:**
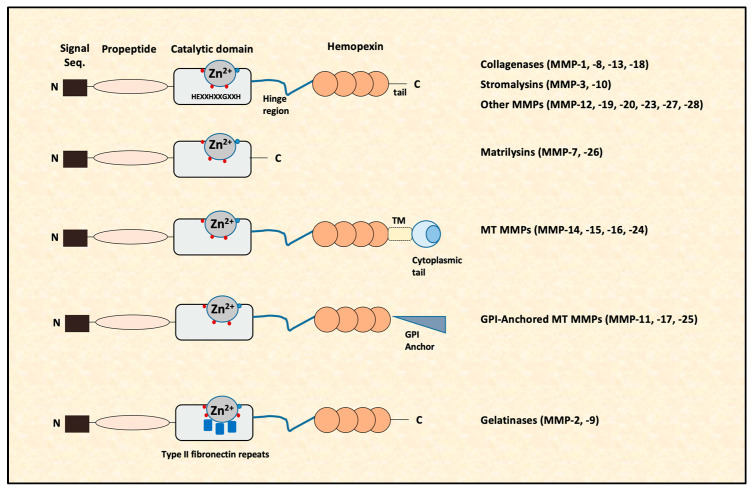
The architecture of matrix metalloproteinases (MMPs). MMPs consist of an N-terminal signal peptide with variable length, a latency-maintaining pro domain, a catalytic domain with a Zn^2+^, a linker-“hinge” region and a C-terminal domain. In all MMPs, apart from MMP-7 and MMP-26, there is a hemopexin-like domain at the C-terminus. In transmembrane MMPs, there is a transmembrane domain (TM) and a short cytoplasmic domain or a glycosylphosphatidylinositol (GPI) anchor that binds them to the cell surface. Gelatinases differ from the other MMPs, as in their catalytic domain, they contain cysteine repeats that resemble collagen binding sites of type II fibronectin. Red dots represent histidines. Blue squares represent Type II fibronectin repeats. MT: membrane-type.

**Figure 2 ijms-24-03786-f002:**
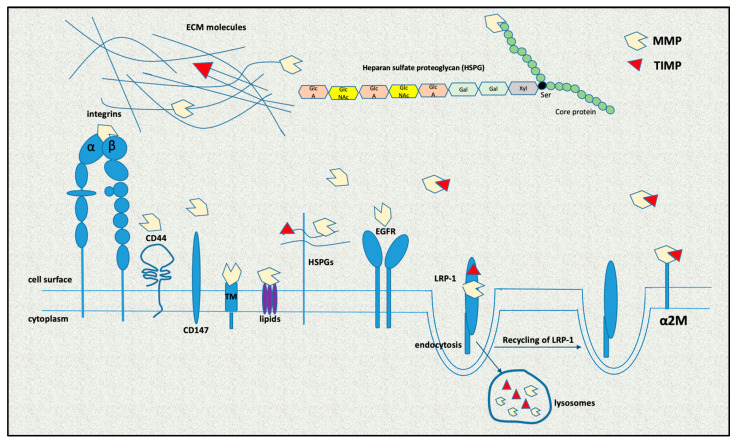
Control of MMP activity. MMPs can be found in every cellular compartment interacting with other proteins; proteoglycan core proteins and/or their glycosaminoglycan chains; extracellular matrix (ECM) molecules; integrins; CD44; CD147; and various receptors, such as epidermal growth factor receptor (EGFR). They may also be subjected to endocytosis by binding to LDL-related protein 1 (LRP-1) receptor. MMP activity is also inhibited by tissue inhibitors of MMPs (TIMPs) and α2-macroglobulin (α2M). TM: transmembrane MMP.

**Figure 3 ijms-24-03786-f003:**
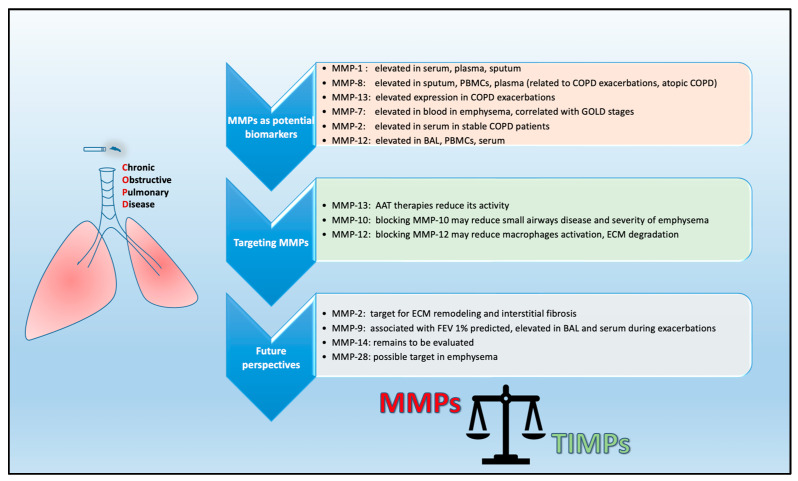
Matrix metalloproteinases (MMPs) in COPD. Various MMPs as potential biomarkers and targets in COPD. BAL: bronchoalveolar lavage; PBMCs: peripheral blood mononuclear cells; AAT: alpha-1 antitrypsin therapy; ECM: extracellular matrix; TIMPs; tissue inhibitors of MMPs.

**Table 1 ijms-24-03786-t001:** Members of the matrix metalloproteinase (MMP) family.

Nomenclature	MMP Nr	Substrate/Action
Collagenases		
Intermediate space collagenase I	MMP-1	Type I, II, III, VII and X collagens; entactin; aggrecan; proteoglycans; β-casein; gelatin; tenascin; myelin basic protein; ovostatin
Collagenase of neutrophils	MMP-8	Type I, II and III collagens; aggrecan; proteoglycans; fibronectin; aggrecan; ovostatin
Collagenase 3	MMP-13	Type I, II, III, IV, IX, X and XIV collagens; tenascin C isoform; laminin; plasminogen; osteonectin; serine protease inhibitors; fibrillin-1; aggrecan core protein
Collagenase 4	MMP-18	Type I collagen, gelatin
Stromalysins		
Stromalysin 1	MMP-3	Aggrecan; fibronectin; laminin; gelatins; type III, IV, IX and X collagens; decorin, myelin; ovastatin; casein; osteonectin; elastin; proteoglycans
Stromalysin 2	MMP-10	Aggrecan; fibronectin; laminin; elastin; type III, IV, V, IX and X collagens; conjugate protein; proteoglycans; carboxymethyl transferrin
Stromalysin 3	MMP-11	Moderate activity against fibrinogen, laminin, type IV collagen, aggrecan, gelatins, serpins, a1 proteinases, a1 antitrypsin inhibitors
Matrilysins		
Matrilysin	MMP-7	Aggrecan; fibronectin; gelatins; type I, II, IV and V collagens; elastin; entactin; syndecan-1; laminin; tenascin; myelin; Faz ligand; pro-TNF-a; E-cadherin
Matrilysin-2 or endometase	MMP-26	(in vitro): type IV collagen, fibronectin, gelatin, vitronectin, a1-antipripsin, b-casein, a2-macrogloboulin
Gelatinases		
Gelatinase A	MMP-2	Gelatins; type I, IV, V, VII, X and XI collagens; fibronectin; laminin; aggrecan; elastin; tenascin; myelin basic protein; vitronectin
Gelatinase B	MMP-9	Gelatins; type IV, V and XI collages; entactin; elastin; aggrecan; cytokines; decorin; casein; chemokines; IL-8; IL-1b; myelin, casein
Membrane-Type		
MT1-MMP	MMP-14	Fibronectin, laminin-1, vitronectin, cartilage proteoglycans, fibrilin-1, tenascin, entactin, aggrecan, a1-proteinase inhibitor, a2-macrogloboulin
MT2-MMP	MMP-15	Laminin, fibronectin, entactin, aggrecan, gelatin, vitronectin, tenascin
MT3-MMP	MMP-16	Gelatin, casein, type III collagen, laminin, fibronectin
MT4-MMP	MMP-17	Gelatin, fibrinogen, fibrin
MT5-MMP	MMP-24	Fibronectin, gelatin, proteoglycans
MT6-MMP (GPI-anchored)	MMP-25	Type IV collagen, fibronectin, gelatin, proteoglycans
Other MMPs		
Macrophage metalloelastase	MMP-12	Gelatin type I; elastin; fibronectin; laminin; vitronectin; proteoglycans; elastin; type I, IV and V collagens; entactin; ostentation; aggrecan; myelin; fibrinogen; a1-antitrypsin
RASI-1	MMP-19	Type I and IV collagens, laminin, nitrogen, tenascin-C isoform, entactin, aggrecan, fibronectin, gelatin type I
Enamelysin	MMP-20	Ameloblasts, aggrecan, odontoblasts, amelogenin
Cysteine array (CA) MMP	MMP-23	Gelatin
	MMP-27	Gelatin
Epilysin	MMP-28	Casein

**Table 2 ijms-24-03786-t002:** Role of MMPs and TIMPs in COPD.

A. Collagenases in COPD	
MMP-1	Elevated in serum of COPD patients and corelates with disease severity sputum levels are increased in smokers and patients with more advanced COPD.
MMP-8	Elevated in sputum of COPD patients; high levels of mRNA and protein expression in PBMCs and plasma from COPD patients and higher expression during exacerbations; detected in exhaled breath condensate of COPD patients; increased in serum of patients with atopic COPD compared with non-atopic COPD; mediates the inflammatory response by inducing neutrophil apoptosis, making COPD patients more susceptible to acute exacerbation due to the effect of allergens.
MMP-13	Upregulated in patients with COPD; involved in COPD exacerbations and elevated in smokers after viral infections; MMP-13-mediated cleavage of α-1 antitrypsin reduce sMMP-13 activity and protects against lung damage; targeting MMP-13 through specific inhibitors or AAT therapies seems beneficial for the treatment of COPD.
**B. Stromalysins in COPD**	
MMP-3	Elevated in COPD patients; high levels contribute to excessive extracellular matrix degradation and worse lung function; serum levels were not found to be elevated in smokers; concentration in BAL is associated with CT markers of small airway disease and related to the severity of emphysema.
MMP-10	Elevated in the lungs of COPD patients; expressed by macrophages and CD68-positive cells in the lungs and, to a lesser extent, by epithelial cells in response to acute inflammatory conditions; macrophage-derived MMP-10 mitigates the proinflammatory response by controlling macrophage activation by restraining M1 polarization and promoting the ability of M2 macrophages to control the expression of gelatinolytic MMPs, particularly MMP-13; its expression is associated with small airway disease and the severity of emphysema.
**C. Matrilysins in COPD**	
MMP-7	Expressed by the mucosal epithelium and macrophages; increased in the blood of COPD patients compared to healthy subjects and associated with deterioration of lung function; an SNP in the promoter of the MMP-7 gene (rs1156818) is associated with a high risk of COPD; serum levels are elevated in patients with emphysema and correlated with GOLD stages.
**D. Gelatinases in COPD**	
MMP-2	Elevated in serum of stable COPD patients compared with asthmatic patients and controls, suggesting MMP-2 as a potential biomarker for COPD; a positive fold change in MMP-2 expression was detected by mass spectrometric analysis of the COPD proteome; local expression of MMP-2, as well as MMP-9 and TIMP-1, was associated with pathological changes in the pulmonary interstitium and lung function of COPD patients, suggesting their involvement in COPD progression; expression and activity were significantly increased in lung tissues after injury and in rats with pulmonary fibrosis; a suitable therapeutic target for COPD pathogenesis
MMP-9	Its expression did not vary in different lung compartments of COPD patients; there was no association either between increased blood MMP-9 expression and progression of emphysema or between MMP-9 mRNA levels in macrophages and markers of ongoing lung injury; COPD patients are prone to produce higher levels of MMP-9 and MMP-9/TIMP-1 complex than healthy individuals, which is correlated with decreased FEV1% levels in patients with COPD compared to controls; the levels of MMP-9 and its ratio relative to TIMP-1 have been associated with increased risk of death; increased levels of MMP-9, TIMP-1 and TIMP-2 were also observed in BAL during acute exacerbations of COPD and were negatively correlated with predicted FEV1%, indicating that MMP-9 and TIMPs may be persistent aggravating factors associated with airway remodeling and obstruction, suggesting a pathway connecting frequent exacerbations to lung function decline; in COPD, it is linked to inflammation and lung remodeling, as it uniquely mediates pulmonary inflammation through ECM degradation, neutrophil chemotaxis and augmentation of inflammation; serum levels, along with PGE2 and COX-2 levels, were found to be enhanced in COPD patients relative to healthy subjects and correlated with GOLD grade, CAT score and clinical history; candidate as a prognostic biomarker for COPD.
**E. Membrane-type MMPs in COPD**	
MMP-14	Cigarette and tobacco smoke extract (TSE) can promote the release of MMP-14; it is involved in emphysema.
**F. Other MMPs in COPD**	
MMP-12	Involvement in alveolar destruction and in the pathogenesis of COPD, acting either as an elastase or affecting macrophage activation; increased expression levels in BAL, in PBMCs and in serum from patients with COPD compared to healthy subjects; the presence of an SNP in the promoter of MMP-12, such as the (-82) A/G allele of SNP rs2276109 or Asn357Ser (A/G) of rs652438, has been associated with reduced risk of developing COPD and better prognosis; acts on elastin degradation.
MMP-28	Plays a causative role in cigarette-smoke-induced emphysema; it is expressed in human COPD lung tissue, while cigarette-smoke-exposed mice have been shown to present increased MMP-28 mRNA levels in both alveolar lung tissue and macrophages; it is associated with stimulation of chemokine expression.
**TIMPS in COPD**	
TIMP-1	A critical regulator of extracellular matrix degradation; plays a key role in limiting inflammation after injury; has the ability to moderate ECM degradation both in healthy tissues and under pathological conditions; preferentially inhibits MMP-9 and may exert protection, while the levels of MMP-9 and the MMP-9/TIMP-1 ratio are reliable predictors of emphysema; TIMP-1 levels were found to be elevated in COPD patients, and plasma TIMP-1 levels were associated with disease severity; TIMP-1 was found to be enhanced in PH, a very common and lethal comorbidity of COPD, compared to COPD patients, suggesting that TIMP-1 levels could be used as a biomarker to identify high-risk patients.
TIMP-2	Involved in abnormal ECM accumulation either directly or indirectly through inhibition of MMP-14 or activation of MMP-2; was found in the bronchial epithelium and connective tissue, along with apparently large numbers of immunoreactive TIMP-2 endothelial cells in COPD patients; plasma levels in COPD patients were higher compared to controls and correlated with plasma MMP-2 levels; the MMP-2/TIMP-2 ratio was elevated in COPD patients and was representative of tobacco use.
TIMP-3	A major inhibitor of ECM remodeling that mediates fibrosis; a comparison of tissues from healthy control subjects and severe cases of COPD GOLD stage IV and end-stage IPF showed that TIMP-3 levels were significantly elevated; a COPD regulator that strongly influences tissue homeostasis.
TIMP-4	Protein levels are significantly upregulated in the serum of COPD patients, along with increased expression of TIMP-4 mRNA in PBMCs.

## Data Availability

Not applicable.

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
