# Peer review of "Matrix Metalloproteinases in Chronic Obstructive Pulmonary Disease"

_ijms, 2023, doi:10.3390/ijms24043786_

Round 1
Reviewer 1 Report
In this review the authors evaluated the role of MMPs in respiratory diseases. The principal authors (first name) has not experience on this topic, while the other authors EP ans DS have a very good skill on both COPD and molecular mechanism.
The manuscript is well written even if there are some points that must be evaluated
1) The type of review is not indicated, please add it
2) Introduction: is very long, please reduce it
3) Methods: please add this section, to understand how you have do the choose of the manuscript
4) Discussion: please add a table to clarify the role of MMPs and TIMP in COPD
5) Section 5: you write: ". In recent years, many studies have been conducted focusing on the discovery 421 of small molecules as potent and selective inhibitors of MMPs (196, 197)." please add also these manuscripts that docuemnted in vitro the role of drugs as MMPs modulators in lung diseases ( doi: 10.1186/2050-6511-15-67.; doi: 10.1111/cpr.12018.)
5)
Reviewer 2 Report
Matrix metalloproteinases in chronic obstructive pulmonary disease
The review article by Christopoulou and colleagues sought to identify the importance and function of matrix metalloproteinases (MMPs) in chronic obstructive pulmonary disease. The authors introduce the background and structure of MMPs and summarize their aberrant events in COPD. The subtypes of MMPs are classified by their definition and function. They also illustrate the regulatory components of MMPs, tissue inhibitors of MMPs. Additionally, they mentioned several MMP inhibitors as potential options for therapeutic strategies. But in this study, they should expand their observations, increase their rigor, and make more constructive suggestions. The following issues must be addressed:
Major comments:
1. In the first paragraph, the authors describe that the MMP contains three histidines that bind Zn2+ at the catalytic site. However, in Figure 1, I only noticed the two histidine residues of the MMP. In addition, what is the full name of GPI? I recommend that they categorize all MMPs into these groups.
2. In line 37, they state that "23 MMPs are synthesized as proenzymes in humans". But they represent 28 family members in Table 1. They need to detail the differences or definitions.
3. The current Table 1 classification is confusing. For example, one category is “Membrane Type”, so what are the other distribution locations? Some MMP families pair with multiple substrates, so is there a priority or affinity?
4. They mention that MMPs may be regulated by epigenetic changes, methylation, or various microRNAs. However, these data are not listed and explained in detail. They need to include events related to COPD.
5. I agree that in COPD, MMPs are abnormally or overexpressed and easily detectable. However, they must take into account some possible regulatory mechanisms. Although due to an inflammatory response, via chemokines, transcription factors and signaling should be described in detail.
6. They assigned MMP-1, -8, -13, -7, -2, and -12 to potential biomarkers. Do they have any prognostic or diagnostic indicators for risk, prevalence or stage of MMP-induced COPD?
7. They describe a few inhibitors in a short space, including AZ11557272, AS11793, and MMP408. In addition to being unable to confirm their function, they were also unable to understand the detailed mechanism.
8. COPD can also be triggered in a variety of ways, including different pathogens, such as certain substances, cigarettes, or chemicals. Are the trends and functions of MMPs/TIMPs similar?
9. They should add some future perspective on how readers and reviewers can focus on MMPs in COPD in further experiments and development.
Minor comments:
1. These figures should be improved and upgraded.
Round 2
Reviewer 1 Report
Dear Authors I have read the manuscript and I have not further comments
Reviewer 2 Report
The authors corrected some typos/errors and improved some weaknesses. I have no further concerns and questions.